# Learning Differentially Private Recurrent Language Models

**H. Brendan McMahan**
mcmahan@google.com

**Daniel Ramage**
dramage@google.com

**Kunal Talwar**
kunal@google.com

**Li Zhang**
liqzhang@google.com

## Abstract

We demonstrate that it is possible to train large recurrent language models with user-level differential privacy guarantees with only a negligible cost in predictive accuracy. Our work builds on recent advances in the training of deep networks on user-partitioned data and privacy accounting for stochastic gradient descent. In particular, we add user-level privacy protection to the federated averaging algorithm, which makes "large step" updates from user-level data. Our work demonstrates that given a dataset with a sufficiently large number of users (a requirement easily met by even small internet-scale datasets), achieving differential privacy comes at the cost of increased computation, rather than in decreased utility as in most prior work. We find that our private LSTM language models are quantitatively and qualitatively similar to un-noised models when trained on a large dataset.

## 1 Introduction

Deep recurrent models like long short-term memory (LSTM) recurrent neural networks (RNNs) have become a standard building block in modern approaches to language modeling, with applications in speech recognition, input decoding for mobile keyboards, and language translation. Because language usage varies widely by problem domain and dataset, training a language model on data from the right distribution is critical. For example, a model to aid typing on a mobile keyboard is better served by training data typed in mobile apps rather than from scanned books or transcribed utterances. However, language data can be uniquely privacy sensitive. In the case of text typed on a mobile phone, this sensitive information might include passwords, text messages, and search queries. In general, language data may identify a speaker—explicitly by name or implicitly, for example via a rare or unique phrase—and link that speaker to secret or sensitive information.

Ideally, a language model's parameters would encode patterns of language use common to many users without memorizing any individual user's unique input sequences. However, we know convolutional NNs can memorize arbitrary labelings of the training data (Zhang et al., 2017) and recurrent language models are also capable of memorizing unique patterns in the training data (Carlini et al., 2018). Recent attacks on neural networks such as those of Shokri et al. (2017) underscore the implicit risk. The main goal of our work is to provide a strong guarantee that the trained model protects the privacy of individuals' data without undue sacrifice in model quality.

We are motivated by the problem of training models for next-word prediction in a mobile keyboard, and use this as a running example. This problem is well suited to the techniques we introduce, as differential privacy may allow for training on data from the true distribution (actual mobile usage) rather than on proxy data from some other source that would produce inferior models. However, to facilitate reproducibility and comparison to non-private models, our experiments are conducted on a public dataset as is standard in differential privacy research. The remainder of this paper is structured around the following contributions:

1. We apply differential privacy to model training using the notion of *user-adjacent* datasets, leading to formal guarantees of user-level privacy, rather than privacy for single examples.

2. We introduce a noised version of the federated averaging algorithm (McMahan et al., 2016) in §2, which satisfies user-adjacent differential privacy via use of the moments accountant (Abadi et al., 2016a) first developed to analyze differentially private stochastic gradient descent (SGD) for example-level privacy. The federated averaging approach groups multiple SGD updates together, enabling large-step model updates.

3. We demonstrate the first high quality LSTM language model trained with strong privacy guarantees in §3, showing no significant decrease in model accuracy given a large enough dataset. For example, on a dataset of 763,430 users, baseline (non-private) training achieves an accuracy of $17.5\%$ in 4120 rounds of training, where we use the data from 100 random users on each round. We achieve this same level of accuracy with $(4.6, 10^{-9})$-differential privacy in 4980 rounds, processing on average 5000 users per round—maintaining the same level of accuracy at a significant computational cost of roughly $60\times$.[1] Running the same computation on a larger dataset with $10^8$ users would improve the privacy guarantee to $(1.2, 10^{-9})$. We guarantee privacy and maintain utility despite the complex internal structure of the LSTM—with per-word embeddings as well as dense state transitions—by using the federated averaging algorithm. We demonstrate that the noised model's metrics and qualitative behavior (with respect to head words) does not differ significantly from the non-private model. To our knowledge, our work represents the most sophisticated machine learning model, judged by the size and the complexity of the model, ever trained with privacy guarantees, and the first such model trained with user-level privacy.

4. In extensive experiments in §3, we offer guidelines for parameter tuning when training complex models with differential privacy guarantees. We show that a small number of experiments can narrow the parameter space into a regime where we pay for privacy not in terms of a loss in utility but in terms of an increased computational cost.

We now introduce a few preliminaries. Differential privacy (DP) (Dwork et al., 2006; Dwork, 2011; Dwork and Roth, 2014) provides a well-tested formalization for the release of information derived from private data. Applied to machine learning, a differentially private training mechanism allows the public release of model parameters with a strong guarantee: adversaries are severely limited in what they can learn about the original training data based on analyzing the parameters, even when they have access to arbitrary side information. Formally, it says:

**Definition 1.** *Differential Privacy: A randomized mechanism $\mathcal{M}: \mathcal{D} \to \mathcal{R}$ with a domain $\mathcal{D}$ (e.g., possible training datasets) and range $\mathcal{R}$ (e.g., all possible trained models) satisfies $(\epsilon, \delta)$-differential privacy if for any two **adjacent** datasets $d, d' \in \mathcal{D}$ and for any subset of outputs $S \subseteq \mathcal{R}$ it holds that $\Pr[\mathcal{M}(d) \in S] \leq e^\epsilon \Pr[\mathcal{M}(d') \in S] + \delta$.*

The definition above leaves open the definition of **adjacent datasets** which will depend on the application. Most prior work on differentially private machine learning (e.g. Chaudhuri et al. (2011); Bassily et al. (2014); Abadi et al. (2016a); Wu et al. (2017); Papernot et al. (2017)) deals with *example-level privacy*: two datasets $d$ and $d'$ are defined to be adjacent if $d'$ can be formed by adding or removing a single training example from $d$. We remark that while the recent PATE approach of (Papernot et al., 2017) can be adapted to give user-level privacy, it is not suited for a language model where the number of classes (possible output words) is large.

For problems like language modeling, protecting individual examples is insufficient—each typed word makes its own contribution to the RNN's training objective, so one user may contribute many thousands of examples to the training data. A sensitive word or phrase may be typed several times by an individual user, but it should still be protected.[2] In this work, we therefore apply the definition of differential privacy to protect whole user histories in the training set. This *user-level privacy* is ensured by using an appropriate adjacency relation:

**Definition 2.** *User-adjacent datasets: Let $d$ and $d'$ be two datasets of training examples, where each example is associated with a user. Then, $d$ and $d'$ are **adjacent** if $d'$ can be formed by adding or removing all of the examples associated with a single user from $d$.*

---

[1]The additional computational cost could be mitigated by initializing by training on a public dataset, rather than starting from random initialization as we do in our experiments.

[2]Differential privacy satisfies a property known as *group privacy* that can allow translation from example-level privacy to user-level privacy at the cost of an increased $\epsilon$. In our setting, such a blackbox approach would incur a prohibitive privacy cost. This forces us to directly address user-level privacy.

Model training that satisfies differential privacy with respect to datasets that are user-adjacent satisfies the intuitive notion of privacy we aim to protect for language modeling: the presence or absence of any specific user's data in the training set has an imperceptible impact on the (distribution over) the parameters of the learned model. It follows that an adversary looking at the trained model cannot infer whether any specific user's data was used in the training, irrespective of what auxiliary information they may have. In particular, differential privacy rules out memorization of sensitive information in a strong information theoretic sense.

## 2 ALGORITHMS FOR USER-LEVEL DIFFERENTIALLY PRIVATE TRAINING

Our private algorithm relies heavily on two prior works: the `FederatedAveraging` (or `FedAvg`) algorithm of McMahan et al. (2016), which trains deep networks on user-partitioned data, and the moments accountant of Abadi et al. (2016a), which provides tight composition guarantees for the repeated application of the Gaussian mechanism combined with amplification-via-sampling. While we have attempted to make the current work as self-contained as possible, the above references provide useful background.

`FedAvg` was introduced by McMahan et al. (2016) for federated learning, where the goal is to train a shared model while leaving the training data on each user's mobile device. Instead, devices download the current model and compute an update by performing local computation on their dataset. It is worthwhile to perform extra computation on each user's data to minimize the number of communication rounds required to train a model, due to the significantly limited bandwidth when training data remains decentralized on mobile devices. We observe, however, that `FedAvg` is of interest even in the datacenter when DP is applied: larger updates are more resistant to noise, and fewer rounds of training can imply less privacy cost. Most importantly, the algorithm naturally forms per-user updates based on a single user's data, and these updates are then averaged to compute the final update applied to the shared model on each round. As we will see, this structure makes it possible to extend the algorithm to provide a user-level differential privacy guarantee.

We also evaluate the `FederatedSGD` algorithm, essentially large-batch SGD where each mini-batch is composed of "microbatches" that include data from a single distinct user. In some datacenter applications `FedSGD` might be preferable to `FedAvg`, since fast networks make it more practical to run more iterations. However, those additional iterations come at a privacy cost. Further, the privacy benefits of federated learning are nicely complementary to those of differential privacy, and `FedAvg` can be applied in the datacenter as well, so we focus on this algorithm while showing that our results also extend to `FedSGD`.

Both `FedAvg` and `FedSGD` are iterative procedures, and in both cases we make the following modifications to the non-private versions in order to achieve differential privacy:

   A) We use random-sized batches where we select users independently with probability $q$, rather than always selecting a fixed number of users.
   B) We enforce clipping of per-user updates so the total update has bounded $L_2$ norm.
   C) We use different estimators for the average update (introduced next).
   D) We add Gaussian noise to the final average update.

The pseudocode for `DP-FedAvg` and `DP-FedSGD` is given as Algorithm 1. In the remainder of this section, we introduce estimators for C) and then different clipping strategies for B). Adding the sampling procedure from A) and noise added in D) allows us to apply the moments accountant to bound the total privacy loss of the algorithm, given in Theorem 1. Finally, we consider the properties of the moments accountant that make training on large datasets particular attractive.

**Bounded-sensitivity estimators for weighted average queries** Randomly sampling users (or training examples) by selecting each independently with probability $q$ is crucial for proving low privacy loss through the use of the moments accountant (Abadi et al., 2016a). However, this procedure produces variable-sized samples $\mathcal{C}$, and when the quantity to be estimated $f(\mathcal{C})$ is an average rather than a sum (as in computing the weighted average update in `FedAvg` or the average loss on a minibatch in SGD with example-level DP), this has ramifications for the sensitivity of the query $f$.

Specifically, we consider weighted databases $d$ where each row $k \in d$ is associated with a particular user, and has an associated weight $w_k \in [0, 1]$. This weight captures the desired influence of the

**Main training loop:**
  *parameters*
    user selection probability $q \in (0, 1]$
    per-user example cap $\hat{w} \in \mathbb{R}^+$
    noise scale $z \in \mathbb{R}^+$
    estimator $\tilde{f}_\mathrm{f}$, or $\tilde{f}_\mathrm{c}$ with param $W_{\min}$
    UserUpdate (for FedAvg or FedSGD)
    ClipFn (FlatClip or PerLayerClip)

  Initialize model $\theta^0$, moments accountant $\mathcal{M}$
  $w_k = \min\left(\frac{n_k}{\hat{w}}, 1\right)$ for all users $k$
  $W = \sum_{k \in d} w_k$
  **for** each round $t = 0, 1, 2, \ldots$ **do**
    $\mathcal{C}^t \leftarrow$ (sample users with probability $q$)
    **for** each user $k \in \mathcal{C}^t$ **in parallel do**
      $\Delta_k^{t+1} \leftarrow$ UserUpdate($k, \theta^t$, ClipFn)
$$\Delta^{t+1} = \begin{cases} \frac{\sum_{k \in \mathcal{C}^t} w_k \Delta_k}{qW} & \text{for } \tilde{f}_\mathrm{f} \\ \frac{\sum_{k \in \mathcal{C}^t} w_k \Delta_k}{\max(qW_{\min}, \sum_{k \in \mathcal{C}^t} w_k)} & \text{for } \tilde{f}_\mathrm{c} \end{cases}$$
    $S \leftarrow$ (bound on $\|\Delta_k\|$ for ClipFn)
    $\sigma \leftarrow \left\{ \frac{zS}{qW} \text{ for } \tilde{f}_\mathrm{f} \text{ or } \frac{2zS}{qW_{\min}} \text{ for } \tilde{f}_\mathrm{c} \right\}$
    $\theta^{t+1} \leftarrow \theta^t + \Delta^{t+1} + \mathcal{N}(0, I\sigma^2)$
    $\mathcal{M}.\texttt{accum\_priv\_spending}(z)$
  print $\mathcal{M}.\texttt{get\_privacy\_spent}()$

**FlatClip($\Delta$):**
  *parameter $S$*
  return $\pi(\Delta, S)$  *// See Eq. (1).*

**PerLayerClip($\Delta$):**
  *parameters $S_1, \ldots S_m$*
  $S = \sqrt{\sum_j S_j^2}$
  **for** each layer $j \in \{1, \ldots, m\}$ **do**
    $\Delta'(j) = \pi(\Delta(j), S_j)$
  return $\Delta'$

**UserUpdateFedAvg($k, \theta^0$, ClipFn):**
  *parameters $B, E, \eta$*
  $\theta \leftarrow \theta^0$
  **for** each local epoch $i$ from 1 to $E$ **do**
    $\mathcal{B} \leftarrow$ ($k$'s data split into size $B$ batches)
    **for** batch $b \in \mathcal{B}$ **do**
      $\theta \leftarrow \theta - \eta \nabla \ell(\theta; b)$
      $\theta \leftarrow \theta^0 + \text{ClipFn}(\theta - \theta^0)$
  return update $\Delta_k = \theta - \theta^0$  *// Already clipped.*

**UserUpdateFedSGD($k, \theta^0$, ClipFn):**
  *parameters $B, \eta$*
  select a batch $b$ of size $B$ from $k$'s examples
  return update $\Delta_k = \text{ClipFn}(-\eta \nabla \ell(\theta; b))$

Algorithm 1: The main loop for `DP-FedAvg` and `DP-FedSGD`, the only difference being in the user update function (UserUpdateFedAvg or UserUpdateFedSGD). The calls on the moments accountant $\mathcal{M}$ refer to the API of Abadi et al. (2016b).

row on the final outcome. For example, we might think of row $k$ containing $n_k$ different training examples all generated by user $k$, with weight $w_k$ proportional to $n_k$. We are then interested in a bounded-sensitivity estimate of $f(\mathcal{C}) = \frac{\sum_{k \in \mathcal{C}} w_k \Delta_k}{\sum_{k \in \mathcal{C}} w_k}$ for per-user vectors $\Delta_k$, for example to estimate the weighted-average user update in `FedAvg`. Let $W = \sum_k w_k$. We consider two such estimators:

$$\tilde{f}_\mathrm{f}(\mathcal{C}) = \frac{\sum_{k \in \mathcal{C}} w_k \Delta_k}{qW}, \qquad \text{and} \qquad \tilde{f}_\mathrm{c}(\mathcal{C}) = \frac{\sum_{k \in \mathcal{C}} w_k \Delta_k}{\max(qW_{\min}, \sum_{k \in \mathcal{C}} w_k)}.$$

Note $\tilde{f}_\mathrm{f}$ is an unbiased estimator, since $\mathbb{E}[\sum_{k \in \mathcal{C}} w_k] = qW$. On the other hand, $\tilde{f}_\mathrm{c}$ matches $f$ exactly as long as we have sufficient weight in the sample. For privacy protection, we need to control the sensitivity of our query function $\tilde{f}$, defined as $\mathbb{S}(\tilde{f}) = \max_{\mathcal{C}, k} \|\tilde{f}(\mathcal{C} \cup \{k\}) - \tilde{f}(\mathcal{C})\|_2$, where the added user $k$ can have arbitrary data. The lower-bound $qW_{\min}$ on the denominator of $\tilde{f}_\mathrm{c}$ is necessary to control sensitivity. Assuming each $w_k \Delta_k$ has bounded norm, we have:

**Lemma 1.** *If for all users $k$ we have $\|w_k \Delta_k\|_2 \leq S$, then the sensitivity of the two estimators is bounded as $\mathbb{S}(\tilde{f}_\mathrm{f}) \leq \frac{S}{qW}$ and $\mathbb{S}(\tilde{f}_\mathrm{c}) \leq \frac{2S}{qW_{min}}$.*

A proof is given in Appendix §A.

**Clipping strategies for multi-layer models** Unfortunately, when the user vectors $\Delta_k$ are gradients (or sums of gradients) from a neural network, we will generally have no a priori bound[3] $S$ such that $\|\Delta_k\| \leq S$. Thus, we will need to "clip" our updates to enforce such a bound before applying $\tilde{f}_\mathrm{f}$ or $\tilde{f}_\mathrm{c}$. For a single vector $\Delta$, we can apply a simple $L_2$ projection when necessary:

$$\pi(\Delta, S) \stackrel{\text{def}}{=} \Delta \cdot \min\left(1, \frac{S}{\|\Delta\|}\right). \tag{1}$$

---

[3]To control sensitivity, Lemma 1 only requires that $\|w_k \Delta_k\|$ is bounded. For simplicity, we only apply clipping to the updates $\Delta_k$, using the fact $w_k \leq 1$, leaving as future work the investigation of weight-aware clipping schemes.

Table 1: Privacy for different total numbers of users $K$ (all with equal weight), expected number of users sampled per round $\tilde{C}$, and the number of rounds of training. For each row, we set $\delta = \frac{1}{K^{1.1}}$ and report the value of $\epsilon$ for which $(\epsilon, \delta)$-differential privacy holds after 1 to $10^6$ rounds. For large datasets, additional rounds of training incur only a minimal additional privacy loss.

| users | sample | noise | Upper bound on privacy $\epsilon$ after $1, 10, \ldots 10^6$ rounds | | | | | | |
|---|---|---|---|---|---|---|---|---|---|
| $K$ | $\tilde{C}$ | $z$ | $10^0$ | $10^1$ | $10^2$ | $10^3$ | $10^4$ | $10^5$ | $10^6$ |
| $10^5$ | $10^2$ | 1.0 | 0.97 | 0.98 | 1.00 | 1.07 | 1.18 | 2.21 | 7.50 |
| $10^6$ | $10^1$ | 1.0 | 0.68 | 0.69 | 0.69 | 0.69 | 0.69 | 0.72 | 0.73 |
| $10^6$ | $10^3$ | 1.0 | 1.17 | 1.17 | 1.20 | 1.28 | 1.39 | 2.44 | 8.13 |
| $10^6$ | $10^4$ | 1.0 | 1.73 | 1.92 | 2.08 | 3.06 | 8.49 | 32.38 | 187.01 |
| $10^6$ | $10^3$ | 3.0 | 0.47 | 0.47 | 0.48 | 0.48 | 0.49 | 0.67 | 1.95 |
| $10^9$ | $10^3$ | 1.0 | 0.84 | 0.84 | 0.84 | 0.85 | 0.88 | 0.88 | 0.88 |

However, for deep networks it is more natural to treat the parameters of each layer as a separate vector. The updates to each of these layers could have vastly different $L_2$ norms, and so it can be preferable to clip each layer separately.

Formally, suppose each update $\Delta_k$ contains $m$ vectors $\Delta_k = (\Delta_k(1), \ldots, \Delta_k(m))$. We consider the following clipping strategies, both of which ensure the total update has norm at most $S$:

1. **Flat clipping** Given an overall clipping parameter $S$, we clip the concatenation of all the layers as $\Delta'_k = \pi(\Delta_k, S)$.
2. **Per-layer clipping** Given a per-layer clipping parameter $S_j$ for each layer, we set $\Delta'_k(j) = \pi(\Delta_k(j), S_j)$. Let $S = \sqrt{\sum_{j=1}^m S_j^2}$. The simplest model-independent choice is to take $S_j = \frac{S}{\sqrt{m}}$ for all $j$, which we use in experiments.

We remark here that clipping itself leads to additional bias, and ideally, we would choose the clipping parameter to be large enough that nearly all updates are smaller than the clip value. On the other hand, a larger $S$ will require more noise in order to achieve privacy, potentially slowing training. We treat $S$ as a hyper-parameter and tune it.

**A privacy guarantee** Once the sensitivity of the chosen estimator is bounded, we may add Gaussian noise scaled to this sensitivity to obtain a privacy guarantee. A simple approach is to use an $(\epsilon, \delta)$-DP bound for this Gaussian mechanism, and apply the privacy amplification lemma and the advanced composition theorem to get a bound on the total privacy cost. We instead use the Moments Accountant of Abadi et al. (2016a) to achieve much tighter privacy bounds. The moments accountant for the sampled Gaussian mechanism upper bounds the total privacy cost of $T$ steps of the Gaussian mechanism with noise $N(0, \sigma^2)$ for $\sigma = z \cdot \mathbb{S}$, where $z$ is a parameter, $\mathbb{S}$ is the sensitivity of the query, and each row is selected with probability $q$. Given a $\delta > 0$, the accountant gives an $\epsilon$ for which this mechanism satisfies $(\epsilon, \delta)$-DP. The following theorem is a slight generalization of the results in Abadi et al. (2016a); see §A for a proof sketch.

**Theorem 1.** *For the estimator $(\tilde{f}_f, \tilde{f}_c)$, the moments accountant of the sampled Gaussian mechanism correctly computes the privacy loss with the noise scale of $z = \sigma/\mathbb{S}$ and steps $T$, where $\mathbb{S} = S/qW$ for $(\tilde{f}_f)$ and $2S/qW_{min}$ for $(\tilde{f}_c)$.*

**Differential privacy for large datasets** We use the implementation of the moments accountant from Abadi et al. (2016b). The moments accountant makes strong use of amplification via sampling, which means increasing dataset size makes achieving high levels of privacy significantly easier. Table 1 summarizes the privacy guarantees offered as we vary some of the key parameters. The takeaway from this table is that as long as we can afford the cost in utility of adding noise proportional to $z$ times the sensitivity of the updates, we can get reasonable privacy guarantees over a large range of parameters. The size of the dataset has a modest impact on the privacy cost of a single query (1 round column), but a large effect on the number of queries that can be run without significantly increasing the privacy cost (compare the $10^6$ round column). For example, on a dataset with $10^9$

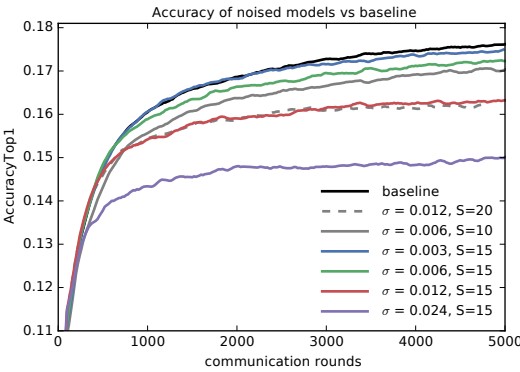

Figure 1: Noised training versus the non-private baseline. The model with $\sigma = 0.003$ nearly matches the baseline.

Table 2: Privacy ($\epsilon$ at $\delta = 10^{-9}$) and accuracy after 5000 rounds of training for models with different $\sigma$ and $S$ from Figure 1. The $\epsilon$'s are strict upper bounds on the true privacy loss given the dataset size $K$ and $\tilde{C}$; AccuracyTop1 (AccT1) is estimated from a model trained with the same $\sigma$ as discussed in the text.

| model | | data | | | |
|---|---|---|---|---|---|
| $\sigma$ | $S$ | users $K$ | $\tilde{C}$ | $\epsilon$ | AccT1 |
| **0.000** | $\infty$ | 763430 | 100 | $\infty$ | 17.62% |
| **0.003** | **15** | 763430 | 5000 | 4.634 | 17.49% |
| **0.006** | **10** | 763430 | 1667 | 2.314 | 17.04% |
| **0.012** | **15** | 763430 | 1250 | 2.038 | 16.33% |
| **0.003** | **15** | $10^8$ | 5000 | 1.152 | 17.49% |
| **0.006** | **10** | $10^8$ | 1667 | 0.991 | 17.04% |
| **0.012** | **15** | $10^8$ | 1250 | 0.987 | 16.33% |

users, the privacy upper bound is nearly constant between 1 and $10^6$ calls to the mechanism (that is, rounds of the optimization algorithm).

There is only a small cost in privacy for increasing the expected number of (equally weighted) users $\tilde{C} = qW = qK$ selected on each round as long as $\tilde{C}$ remains a small fraction of the size of the total dataset. Since the sensitivity of an average query decreases like $1/\tilde{C}$ (and hence the amount of noise we need to add decreases proportionally), we can increase $\tilde{C}$ until we arrive at a noise level that does not adversely effect the optimization process. We show empirically that such a level exists in the experiments.

## 3 EXPERIMENTAL RESULTS

In this section, we evaluate `DP-FedAvg` while training an LSTM RNN tuned for language modeling in a mobile keyboard. We vary noise, clipping, and the number of users per round to develop an intuition of how privacy affects model quality in practice.

We defer our experimental results on `FedSGD` as well as on models with larger dictionaries to Appendix §D. To summarize, they show that `FedAvg` gives better privacy-utility trade-offs than `FedSGD`, and that our empirical conclusions extend to larger dictionaries with relatively little need for additional parameter tuning despite the significantly larger models. Some less important plots are deferred to §C.

**Model structure** The goal of a language model is to predict the next word in a sequence $s_t$ from the preceding words $s_0...s_{t-1}$. The neural language model architecture used here is a variant of the LSTM recurrent neural network (Hochreiter and Schmidhuber, 1997) trained to predict the next word (from a fixed dictionary) given the current word and a state vector passed from the previous time step. LSTM language models are competitive with traditional n-gram models (Sundermeyer et al., 2012) and are a standard baseline for a variety of ever more advanced neural language model architectures (Grave et al., 2016; Merity et al., 2016; Gal and Ghahramani, 2016). Our model uses a few tricks to decrease the size for deployment on mobile devices (total size is 1.35M parameters), but is otherwise standard. We evaluate using `AccuracyTop1`, the probability that the word to which the model assigns highest probability is correct . Details on the model and evaluation metrics are given in §B. All training began from a common random initialization, though for real-world applications pre-training on public data is likely preferable (see §B for additional discussion).

**Dataset** We use a large public dataset of Reddit posts, as described by Al-Rfou et al. (2016). Critically for our purposes, each post in the database is keyed by an author, so we can group the data by these keys in order to provide user-level privacy. We preprocessed the dataset to $K = 763,430$ users each with 1600 tokens. Thus, we take $w_k = 1$ for all users, so $W = K$. We write $\tilde{C} = $

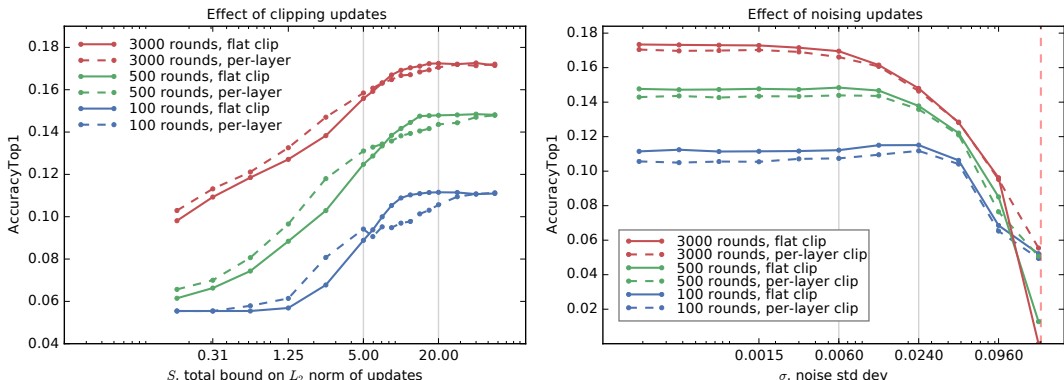

Figure 2: The effect of update clipping on the convergence of `FedAvg`, after 100, 500, and 3000 rounds of training.

Figure 3: The effect of different levels of noise $\sigma$ for flat and per-layer clipping at $S = 20$. The vertical dashed red line is $\sigma = 0.2$.

$qK = qW$ for the expected number of users sampled per round. See §B for details on the dataset and preprocessing. To allow for frequent evaluation, we use a relatively small test set of 75122 tokens formed from random held-out posts. We evaluate accuracy every 20 rounds and plot metrics smoothed over 5 evaluations (100 rounds).

**Building towards DP: sampling, estimators, clipping, and noise**   Recall achieving differential privacy for `FedAvg` required a number of changes (§2, items A–D). In this section, we examine the impact of each of these changes, both to understand the immediate effects and to enable the selection of reasonable parameters for our final DP experiments. This sequence of experiments also provides a general road-map for applying differentially private training to new models and datasets. For these experiments, we use the `FedAvg` algorithm with a fixed learning rate of 6.0, which we verified was a reasonable choice in preliminary experiments.[4] In all `FedAvg` experiments, we used a local batch size of $B = 8$, an unroll size of 10 tokens, and made $E = 1$ passes over the local dataset; thus `FedAvg` processes 80 tokens per batch, processing a user's 1600 tokens in 20 batches per round.

First, we investigate the impact of changing the estimator used for the average per-round update, as well as replacing a fixed sample of $C = 100$ users per round to a variable-sized sample formed by selecting each user with probability $q = 100/763430$ for an expectation of $\tilde{C} = 100$ users. None of these changes significantly impacted the convergence rate of the algorithm (see Figure 5 in §C). In particular, the fixed denominator estimator $\tilde{f}_\mathrm{f}$ works just as well as the higher-sensitivity clipped-denominator estimator $\tilde{f}_\mathrm{c}$. Thus, in the remaining experiments we focus on estimator $\tilde{f}_\mathrm{f}$.

Next, we investigate the impact of flat and per-layer clipping on the convergence rate of `FedAvg`. The model has 11 parameter vectors, and for per-layer clipping we simply chose to distribute the clipping budget equally across layers with $S_j = S/\sqrt{11}$. Figure 2 shows that choosing $S \in [10, 20]$ has at most a small effect on convergence rate.

Finally, Figure 3 shows the impact of various levels of per-coordinate Gaussian noise $\mathcal{N}(0, \sigma^2)$ added to the average update. Early in training, we see almost no loss in convergence for a noise of $\sigma = 0.024$; later in training noise has a larger effect, and we see a small decrease in convergence past $\sigma = 0.012$. These experiments, where we sample only an expected 100 users per round, are not sufficient to provide a meaningful privacy guarantee. We have $S = 20.0$ and $\tilde{C} = qW = 100$, so the sensitivity of estimator $\tilde{f}_\mathrm{f}$ is $20/100.0 = 0.2$. Thus, to use the moments accountant with $z = 1$, we would need to add noise $\sigma = 0.2$ (dashed red vertical line), which destroys accuracy.

**Estimating the accuracy of private models for large datasets**   Continuing the above example, if instead we choose $q$ so $\tilde{C} = 1250$, set the $L_2$ norm bound $S = 15.0$, then we have sensitivity

---

[4]The proper choice of for the clipping parameters may depend on the learning rate, so if the learning rate is changed, clipping parameter choices will also need to be re-evaluated.

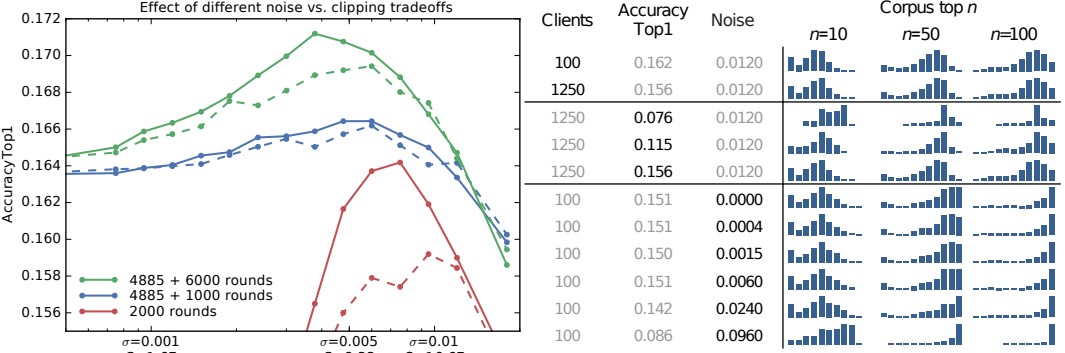

Figure 4: Different noise/clipping tradeoffs (all of equal privacy cost), for initial training (red) and adjusted after 4885 rounds (green and blue). Solid lines use flat clipping, dashed are per-layer.

Table 3: Count histograms recording how many of a model's (row's) top 10 predictions are found in the $n = 10$, 50, or 100 most frequent words in the corpus. Models that predict corpus top-$n$ more frequently have more mass to the right.

$15/1250 = 0.012$, and so we add noise $\sigma = 0.012$ and can apply the moments account with noise scale $z = 1$. The computation is now significantly more computationally expensive, but will give a guarantee of $(1.97, 10^{-9})$-differential privacy after 3000 rounds of training. Because running such experiments is so computationally expensive, for experimental purposes it is useful to ask: does using an expected 1250 users per round produce a model with different *accuracy* than a model trained with only 100 expected users per round? If the answer is no, we can train a model with $\tilde{C} = 100$ and a particular noise level $\sigma$, and use that model to estimate the utility of a model trained with a much larger $q$ (and hence a much better privacy guarantee). We can then run the moments accountant (without actually training) to numerically upper bound the privacy loss. To test this, we trained two models, both with $S = 15$ and $\sigma = 0.012$, one with $\tilde{C} = 100$ and one with $\tilde{C} = 1250$; recall the first model achieves a vacuous privacy guarantee, while the second achieves $(1.97, 10^{-9})$-differential privacy after 3000 rounds. Figure 7 in §C shows the two models produce almost identical accuracy curves during training. Using this observation, we can use the accuracy of models trained with $\tilde{C} = 100$ to estimate the utility of private models trained with much larger $\tilde{C}$. See also Figure 6 in §C, which also shows diminishing returns for larger $C$ for the standard `FedAvg` algorithm.

Figure 1 compares the true-average fixed-sample baseline model (see Figure 5 in §C) with models that use varying levels of clipping $S$ and noise $\sigma$ at $\tilde{C} = 100$. Using the above approach, we can use these experiments to estimate the utility of LSTMs trained with differential privacy for different sized datasets and different values of $\tilde{C}$. Table 2 shows representative values setting $\tilde{C}$ so that $z = 1$. For example, the model with $\sigma = 0.003$ and $S = 15$ is only worse than the baseline by an additive $-0.13\%$ in AccuracyTop1 and achieves $(4.6, 10^{-9})$-differential privacy when trained with $\tilde{C} = 5000$ expected users per round. As a point of comparison, we have observed that training on a different corpus can cost an additive $-2.50\%$ in AccuracyTop1.[5]

**Adjusting noise and clipping as training progresses** Figure 1 shows that as training progresses, each level of noise eventually becomes detrimental (the line drops somewhat below the baseline). This suggests using a smaller $\sigma$ and correspondingly smaller $S$ (thus fixing $z$ so the privacy cost of each round is unchanged) as training progresses. Figure 4 (and Figure 8 in §C) shows this can be effective. We indeed observe that early in training (red), $S$ in the $10 - 12.6$ range works well ($\sigma = 0.006 - 0.0076$). However, if we adjust the clipping/noise tradeoff after 4885 rounds of training and continue for another 6000, switching to $S = 7.9$ and $\sigma = 0.0048$ performs better.

**Comparing DP and non-DP models** While noised training with `DP-FedAvg` has only a small effect on predictive accuracy, it could still have a large qualitative effect on predictions. We hy-

---

[5]This experiment was performed on different datasets, comparing training on a dataset of public social media posts to training on a proprietary dataset which is more representative of mobile keyboard usage, and evaluating on a held-out sample of the same representative dataset. Absolute AccuracyTop1 was similar to the values we report here for the Reddit dataset.

pothesized that noising updates might bias the model away from rarer words (whose embeddings get less frequent actual updates and hence are potentially more influenced by noise) and toward the common "head" words. To evaluate this hypothesis, we computed predictions on a sample of the test set using a variety of models. At each $s_t$ we intersect the top 10 predictions with the most frequent $10, 50, 100$ words in the dictionary. So for example, an intersection of size two in the top 50 means two of the model's top 10 predictions are in the 50 most common words in the dictionary. Table 3 gives histograms of these counts. We find that better models (higher AccuracyTop1) tend to use fewer head words, but see little difference from changing $\tilde{C}$ or the noise $\sigma$ (until, that is, enough noise has been added to compromise model quality, at which point the degraded model's bias toward the head matches models of similar quality with less noise).

## 4  CONCLUSIONS

In this work, we introduced an algorithm for user-level differentially private training of large neural networks, in particular a complex sequence model for next-word prediction. We empirically evaluated the algorithm on a realistic dataset and demonstrated that such training is possible at a negligible loss in utility, instead paying a cost in additional computation. Such private training, combined with federated learning (which leaves the sensitive training data on device rather than centralizing it), shows the possibility of training models with significant privacy guarantees for important real world applications. Much future work remains, for example designing private algorithms that automate and make adaptive the tuning of the clipping/noise tradeoff, and the application to a wider range of model families and architectures, for example GRUs and character-level models. Our work also highlights the open direction of reducing the computational overhead of differentially private training of non-convex models.

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

## A   ADDITIONAL PROOFS

*Proof of Lemma 1.* For the first bound, observe the numerator in the estimator $\tilde{f}_{\mathrm{f}}$ can change by at most $S$ between neighboring databases, by assumption. The denominator is a constant. For the second bound, the estimator $\tilde{f}_{\mathrm{c}}$ can be thought of as the sum of the vectors $w_k \Delta_k$ divided by $\max(qW_{min}, \sum_{k \in \mathcal{C}} \Delta_k)$. Writing $\mathrm{Num}(\mathcal{C})$ for the numerator $\sum_{k \in \mathcal{C}} w_k \Delta_k$, and $\mathrm{Den}(\mathcal{C})$ for the denominator $\max(qW_{\min}, \sum_{k \in \mathcal{C}} w_k)$, the following are immediate for any $\mathcal{C}$ and $\mathcal{C}' \stackrel{\text{def}}{=} \mathcal{C} \cup \{k\}$:

$$\| \mathrm{Num}(\mathcal{C}') - \mathrm{Num}(\mathcal{C}) \| = \| w_k \Delta_k \| \leq S.$$
$$\| \mathrm{Den}(\mathcal{C}') - \mathrm{Den}(\mathcal{C}) \| \leq 1.$$
$$\| \mathrm{Den}(\mathcal{C}') \| \geq qW_{\min}.$$

It follows that

$$
\begin{aligned}
\| \tilde{f}_{\mathrm{c}}(\mathcal{C}') - \tilde{f}_{\mathrm{c}}(\mathcal{C}) \| &= \left\| \frac{\mathrm{Num}(\mathcal{C}')}{\mathrm{Den}(\mathcal{C}')} - \frac{\mathrm{Num}(\mathcal{C})}{\mathrm{Den}(\mathcal{C})} \right\| \\
&= \left\| \frac{\mathrm{Num}(\mathcal{C}') - \mathrm{Num}(\mathcal{C})}{\mathrm{Den}(\mathcal{C}')} + \mathrm{Num}(\mathcal{C}) \left( \frac{1}{\mathrm{Den}(\mathcal{C}')} - \frac{1}{\mathrm{Den}(\mathcal{C})} \right) \right\| \\
&\leq \left\| \frac{w_k \Delta_k}{\mathrm{Den}(\mathcal{C}')} \right\| + \left\| \frac{\mathrm{Num}(\mathcal{C})}{\mathrm{Den}(\mathcal{C})} \left( \frac{\mathrm{Den}(\mathcal{C}) - \mathrm{Den}(\mathcal{C}')}{\mathrm{Den}(\mathcal{C}')} \right) \right\| \\
&\leq \frac{S}{qW_{\min}} + \| \tilde{f}_{\mathrm{c}}(\mathcal{C}) \| \left( \frac{1}{qW_{\min}} \right) \\
&\leq \frac{2S}{qW_{\min}}.
\end{aligned}
$$

Here in the last step, we used the fact that $\| \tilde{f}_{\mathrm{c}}(\mathcal{C}) \| \leq S$. The claim follows.  □

*Proof of Theorem 1.* It suffices to verify that 1. the moments (of the privacy loss) at each step are correctly bounded; and, 2. the composability holds when accumulating the moments of multiple steps.

At each step, users are selected randomly with probability $q$. If in addition the $L_2$-norm of each user's update is upper-bounded by $\mathbb{S}$, then the moments can be upper-bounded by that of the sampled Gaussian mechanism with sensitivity 1, noise scale $\sigma / \mathbb{S}$, and sampling probability $q$.

Our algorithm, as described in Figure 1, uses a fixed noise variance and generates the i.i.d. noise independent of the private data. Hence we can apply the composability as in Theorem 2.1 in Abadi et al. (2016a).

We obtain the theorem by combining the above and the sensitivity bounds $\tilde{f}_{\mathrm{f}}$ and $\tilde{f}_{\mathrm{c}}$.  □

## B   EXPERIMENT DETAILS

**Model**   The first step in training a word-level recurrent language model is selecting the vocabulary of words to model, with remaining words mapped to a special "UNK" (unknown) token. Training a fully differentially private language model from scratch requires a private mechanism to discover which words are frequent across the corpus, for example using techniques like distributed heavy-hitter estimation (Chan et al., 2012; Bassily et al., 2017). For this work, we simplified the problem by pre-selecting a dictionary of the most frequent 10,000 words (after normalization) in a large corpus of mixed material from the web and message boards (but not our training or test dataset).

Our recurrent language model works as follows: word $s_t$ is mapped to an embedding vector $e_t \in \mathbb{R}^{96}$ by looking up the word in the model's vocabulary. The $e_t$ is composed with the state emitted by the model in the previous time step $s_{t-1} \in \mathbb{R}^{256}$ to emit a new state vector $s_t$ and an "output embedding" $o_t \in \mathbb{R}^{96}$. The details of how the LSTM composes $e_t$ and $s_{t-1}$ can be found in Hochreiter and Schmidhuber (1997). The output embedding is scored against the embedding of each item in the vocabulary via inner product, before being normalized via softmax to compute a probability distribution over the vocabulary. Like other standard language modeling applications,

we treat every input sequence as beginning with an implicit "BOS" (beginning of sequence) token and ending with an implicit "EOS" (end of sequence) token.

Unlike standard LSTM language models, our model uses the same learned embedding for the input tokens and for determining the predicted distribution on output tokens from the softmax.[6] This reduces the size of the model by about 40% for a small decrease in model quality, an advantageous tradeoff for mobile applications. Another change from many standard LSTM RNN approaches is that we train these models to restrict the word embeddings to have a fixed $L_2$ norm of 1.0, a modification found in earlier experiments to improve convergence time. In total the model has 1.35M trainable parameters.

**Initialization and personalization** For many applications public proxy data is available, e.g., for next-word prediction one could use public domain books, Wikipedia articles, or other web content. In this case, an initial model trained with standard (non-private) algorithms on the public data (which is likely drawn from the wrong distribution) can then be further refined by continuing with differentially-private training on the private data for the precise problem at hand. Such pre-training is likely the best approach for practical applications. However, since training models purely on private data (starting from random initialization) is a strictly harder problem, we focus on this scenario for our experiments.

Our focus is also on training a single model which is shared by all users. However, we note that our approach is fully compatible with further on-device personalization of these models to the particular data of each user. It is also possible to give the central model some ability to personalize simply by providing information about the user as a feature vector along with the raw text input. LSTMs are well-suited to incorporating such additional context.

**Accuracy metrics** We evaluate using `AccuracyTop1`, the probability that the word to which the model assigns highest probability is correct (after some minimal normalization). We always count it as a mistake if the true next word is not in the dictionary, even if the model predicts UNK, in order to allow fair comparisons of models using different dictionaries. In our experiments, we found that our model architecture is competitive on `AccuracyTop1` and related metrics (Top3, Top5, and perplexity) across a variety of tasks and corpora.

**Dataset** The Reddit dataset can be accessed through Google BigQuery (Reddit Comments Dataset). Since our goal is to limit the contribution of any one author to the final model, it is not necessary to include all the data from users with a large number of posts. On the other hand, processing users with too little data slows experiments (due to constant per-user overhead). Thus, we use a training set where we have removed all users with fewer than 1600 tokens (words), and truncated the remaining $K = 763,430$ users to have exactly 1600 tokens.

We intentionally chose a public dataset for research purposes, but carefully chose one with a structure and contents similar to private datasets that arise in real-world language modeling task such as predicting the next-word in a mobile keyboard. This allows for reproducibility, comparisons to non-private models, and inspection of the data to understand the impact of differential privacy beyond coarse aggregate statistics (as in Table 3).

---

[6]Press and Wolf (2017) independently introduced this technique and provide an empirical analysis comparing models with and without weight tying.

# C  SUPPLEMENTARY PLOTS

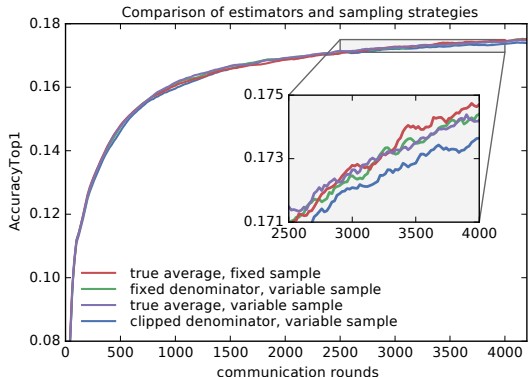

Figure 5: Comparison of sampling strategies and estimators. *Fixed sample* is exactly $C = 100$ users per round, and *variable sample* selects uniformly with probability $q$ for $\tilde{C} = 100$. The *true average* corresponds to $f$, *fixed denominator* is $\tilde{f}_{\mathrm{f}}$, and *clipped denominator* is $\tilde{f}_{\mathrm{c}}$.

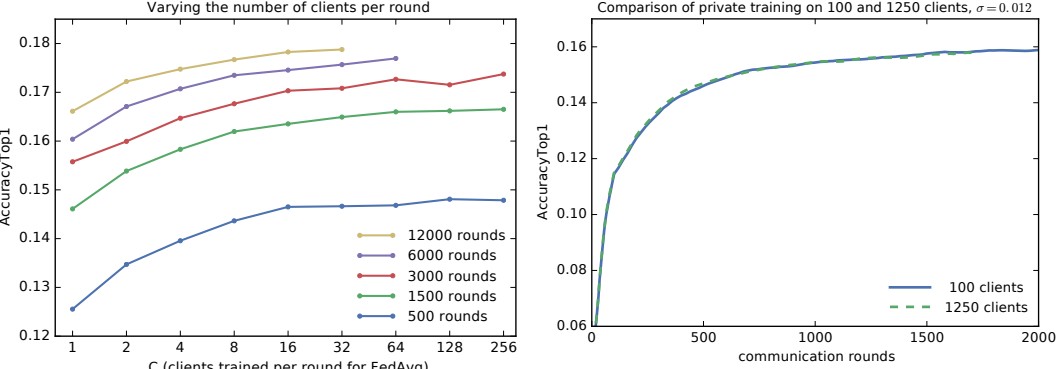

Figure 6: The effect of $C$ for `FedAvg` using the exact estimator and without noise or clipping.

Figure 7: Training with (expected) 100 vs 1250 users per round, both with flat-clipping at $S = 15.0$ and per-coordinate noise with $\sigma = 0.012$.

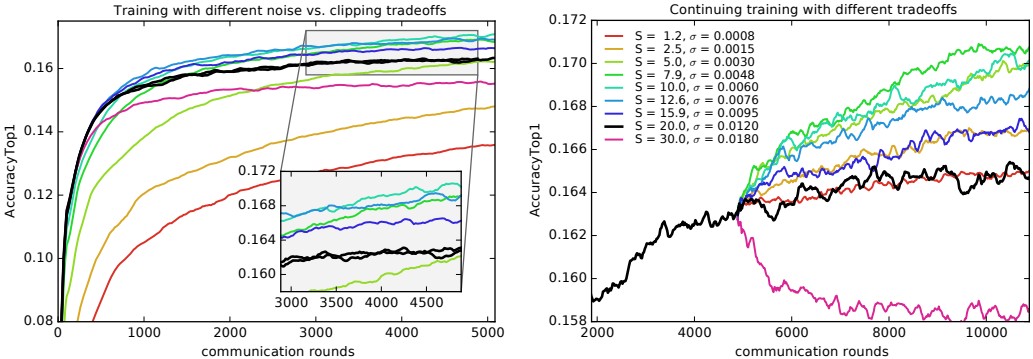

Figure 8: The effect of different noise vs. clipping tradeoffs on convergence. Both plots use the same legend, where we vary $S$ and $\sigma$ together to maintain the same $z = 0.06$ with 100 users (actually used), or $z = 1$ with 1667 users. We take $S = 20$ and $\sigma = 0.012$ (black line) as a baseline; the left-hand plot shows training from a randomly initialized model, and includes two different runs with $S = 20$, showing only mild variability. For the right-hand plot, we took a snapshot of the $S = 20$ model after 4885 initial rounds of training, and resumed training with different tradeoffs.

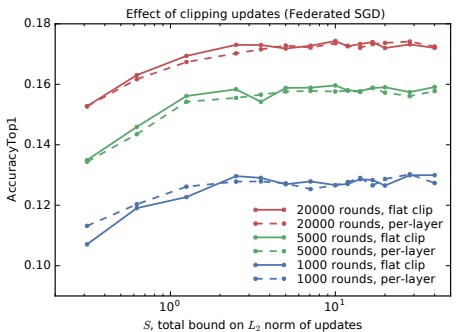

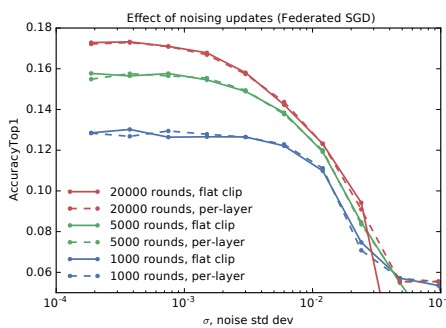

Figure 9: Effect of clipping on `FedSGD` with $\tilde{C} = 50$ users per round and a learning rate of $\eta = 6$. A much smaller clipping level $S$ can be used compared to `FedAvg`.

Figure 10: Effect of noised updates on `FedSGD` with $S = 20$ (based on Figure 9, a smaller value would actually be better when doing private training). `FedSGD` is more sensitive to noise than `FedAvg`, likely because the updates are smaller in magnitude.

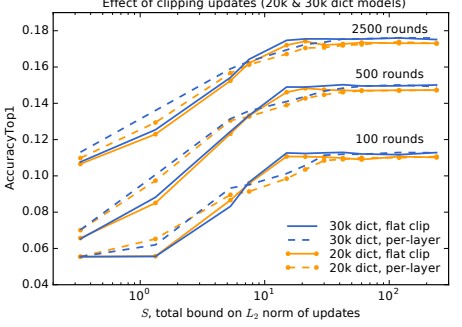

Figure 11: Effect of clipping on models with larger dictionaries (20000 and 30000 tokens).

Figure 12: Effect of noised updates on models with larger dictionaries, when clipped at $S = 20$.

## D  ADDITIONAL EXPERIMENTS

**Experiments with SGD**   We ran experiments using `FedSGD` taking $B = 1600$, that is, computing the gradient on each user's full local dataset. To allow more iterations, we used $\tilde{C} = 50$ rather than 100. Examining Figures 9 and 10, we see $S = 2$ and $\sigma = 2 \cdot 10^{-3}$ are reasonable values, which suggests for private training we would need in expectation $qW = S/\sigma = 1500$ users per round, whereas for `FedAvg` we might choose $S = 15$ and $\sigma = 10^{-2}$ for $\tilde{C} = qW = 1000$ users per round. That is, the relative effect of the ratio of the clipping level to noise is similar between `FedAvg` and `FedSGD`. However, `FedSGD` takes a significantly larger number of iterations to reach equivalent accuracy. Fixing $z = 1$, $\tilde{C} = 5000$ (the value that produced the best accuracy for a private model in Table 2) and total of 763,430 users gives $(3.81, 10^{-9})$-DP after 3000 rounds and $(8.92, 10^{-9})$-DP after 20000 rounds, so there is indeed a significant cost in privacy to these additional iterations.

**Models with larger dictionaries**   We repeated experiments on the impact of clipping and noise on models with 20000 and 30000 token dictionaries, again using `FedAvg` training with $\eta = 6$, equally weighted users with 1600 tokens, and $\tilde{C} = 100$ expected users per round. The larger dictionaries give only a modest improvement in accuracy, and do not require changing the clipping and noise parameters despite having significantly more parameters. Results are given in Figures 11 and 12.

**Other experiments**   We experimented with adding an explicit $L_2$ penalty on the model updates (not the full model) on each user, hoping this would decrease the need for clipping by preferring updates with a smaller $L_2$ norm. However, we saw no positive effect from this.

