# OpenReview forum: "Learning Differentially Private Recurrent Language Models"
_ICLR.cc/2018/Conference — Accept (Poster)_

### Official Review · AnonReviewer2 · 2017-11-27
**I like the experimental strength of the paper, but I have mild concerns about the new algorithmic ideas in the paper.**

**Rating:** 7
**Confidence:** 4

**Review:**

Summary: The paper provides the first evidence of effectively training large RNN based language models under the constraint of differential privacy. The paper focuses on the user-level privacy setting, where the complete contribution of a single user is protected as opposed to protecting a single training example. The algorithm is based on the Federated Averaging and Federated Stochastic gradient framework.

Positive aspects of the paper: The paper is a very strong empirical paper, with experiments comparable to industrial scale. The paper uses the right composition tools like moments accountant to get strong privacy guarantees. The main technical ideas in the paper seem to be i) bounding the sensitivity for weighted average queries, and ii) clipping strategies for the gradient parameters, in order to control the norm. Both these contributions are important in the effectiveness of the overall algorithm.

Concern: The paper seems to be focused on demonstrating the effectiveness of previous approaches to the setting of language models. I did not find strong algorithmic ideas in the paper. I found the paper to be lacking in that respect.

---

### Official Review · AnonReviewer1 · 2017-11-27
**Nice extensions to FederatedAveraging, with strong experimental setup.**

**Rating:** 7
**Confidence:** 2

**Review:**


Summary of the paper
-------------------------------

The authors propose to add 4 elements to the 'FederatedAveraging' algorithm to provide a user-level differential privacy guarantee. The impact of those 4 elements on the model'a accuracy and privacy is then carefully analysed.

Clarity, Significance and Correctness
--------------------------------------------------

Clarity: Excellent

Significance: I'm not familiar with the literature of differential privacy, so I'll let more knowledgeable reviewers evaluate this point.

Correctness: The paper is technically correct.

Questions
--------------

1. Figure 1: Adding some noise to the updates could be view as some form of regularization, so I have trouble understand why the models with noise are less efficient than the baseline.
2. Clipping is supposed to help with the exploding gradients problem. Do you have an idea why a low threshold hurts the performances? Is it because it reduces the amplitude of the updates (and thus simply slows down the training)?
3. Is your method compatible with other optimizers, such as RMSprop or ADAM (which are commonly used to train RNNs)?

Pros
------

1. Nice extensions to FederatedAveraging to provide privacy guarantee.
2. Strong experimental setup that analyses in details the proposed extensions.
3. Experiments performed on public datasets.

Cons
-------

None

Typos
--------

1. Section 2, paragraph 3 : "is given in Figure 1" -> "is given in Algorithm 1"

Note
-------

Since I'm not familiar with the differential privacy literature, I'm flexible with my evaluation based on what other reviewers with more expertise have to say.

---

> ### Author Response · Authors · 2017-12-15
> **Author response**
>
> We thank the reviewer for the thoughtful review and good questions, which we address below:
>
> 1. Figure 1: Adding some noise to the updates could be view as some form of regularization, so I have trouble understand why the models with noise are less efficient than the baseline.
>
> Indeed, we were hoping to see some regularization benefit from noise, but there does not appear to be a significant effect, at least for these models. In Figure 3, which isolates the noise addition, we do see a slight improvement with a modest amount of noise early in training (blue line, noise around 0.012), but otherwise the Gaussian noise we add does not appear to help. We did not do training set evaluation on these models, it is possible (and likely based on results from "Deep learning with differential privacy", Figs. 3 and 6) that the addition of noise decreases the gap between test and training accuracy. Other work has also observed that adding noise may not work well as a regularizer for LSTMs, see  the "negative results" paragraph in Sec 4 of https://openreview.net/pdf?id=rkjZ2Pcxe
>
> 2. Clipping is supposed to help with the exploding gradients problem. Do you have an idea why a low threshold hurts the performances? Is it because it reduces the amplitude of the updates (and thus simply slows down the training)?
>
> This is an important direction for future work, but we have some preliminary thoughts. First, to clarify, note we are clipping each user's update before averaging across users, whereas traditional clipping is applied to a single minibatch update after averaging over examples, and so it is possible that these two types of clipping behave differently.
>
> We suspect two primary reasons for the drop in performance with over-aggressive clipping: (1) reduction in the amplitude of the updates, as you suggest; and (2) clipping introduces bias into the way updates from different users are weighted, essentially changing the loss function being optimized. Some preliminary subsequent experiments indicate that both effects are significant, and that the effect of (1) can be somewhat offset by rescaling the updates on the server. Nevertheless, we emphasize that our primary result is that despite these effects, it is possible to set the clipping parameter large enough that we can still train high-accuracy models.
>
> 3. Is your method compatible with other optimizers, such as RMSprop or ADAM (which are commonly used to train RNNs)?
>
> There are multiple ways these optimizers could be extended to the federated setting. Either algorithm could be applied locally on each client (that is, inside UserUpdateFedAvg) to compute the update, and our approach would work without modification. Running these algorithms across clients while combining them with the additional local computation done by FederatedAverging (which we found to be important for achieving DP) would essentially mean designing a new optimization procedure --- certainly an interesting direction, but beyond the scope of the current work.

---

> > ### Comment · AnonReviewer1 · 2018-01-11
> > **Thanks for your response!**
> >
> >
> > 1. Noise
> >
> > Thanks for the reference. It might indeed be an LSTM issue!
> >
> > 2. Clipping
> >
> > Oh right, I didn't thought about the bias introduces, that is a good point!
> >
> > 3.  Optimizers
> > "Certainly an interesting direction, but beyond the scope of the current work."
> >
> > Indeed!

---

### Official Review · AnonReviewer3 · 2017-11-27
**Nice work**

**Rating:** 8
**Confidence:** 4

**Review:**

This paper extends the previous results on differentially private SGD to user-level differentially private recurrent language models. It experimentally shows that the proposed differentially private LSTM achieves comparable utility compared to the non-private model.

The idea of training differentially private neural network is interesting and very important to the machine learning + differential privacy community. This work makes a pretty significant contribution to such topic. It adapts techniques from some previous work to address the difficulties in training language model and providing user-level privacy. The experiment shows good privacy and utility.

The presentation of the paper can be improved a bit. For example, it might be better to have a preliminary section before Section2 introducing the original differentially private SGD algorithm with clipping, the original FedAvg and FedSGD, and moments accountant as well as privacy amplification; otherwise, it can be pretty difficult for readers who are not familiar with those concepts to fully understand the paper. Such introduction can also help readers understand the difficulty of adapting the original algorithms and appreciate the contributions of this work.

---

> ### Author Response · Authors · 2017-12-15
> **Author response**
>
> We thank the reviewer for the thoughtful review, and will attempt to improve the presentation in the final version. While it will be difficult to fit a complete introduction of all the topics mentioned into the page limit, we will add additional coverage of this material.

---

### Decision · Program_Chairs · 2018-01-29
**ICLR 2018 Conference Acceptance Decision**

**Decision:**

Accept (Poster)

**Comment:**

This paper uses known methods for learning a differentially private models and applies it to the task of learning a language model, and find they are able to maintain accuracy results on large datasets. Reviewers found the method convincing and original saying it was "interesting and very important to the machine learning ... community", and that in terms of results it was a "very strong empirical paper, with experiments comparable to industrial scale". There were some complaints as to the clarity of the work, with requests for more clear explanations of the methods used.